# Incidence of Dysphagia and Comorbidities in Patients with Cervical Dystonia, Analyzed by Botulinum Neurotoxin Treatment Exposure

**DOI:** 10.3390/toxins17030148

**Published:** 2025-03-19

**Authors:** Richard L. Barbano, Bahman Jabbari, Marjan Sadeghi, Ahunna Ukah, Emma Yue, Kimberly Becker Ifantides, Nuo-Yu Huang, David Swope

**Affiliations:** 1Department of Neurology, University of Rochester Medical Center, Rochester, NY 14642, USA; richard_barbano@urmc.rochester.edu; 2Department of Neurology, Yale School of Medicine, New Haven, CT 06510, USA; bahman.jabbari@yale.edu; 3AbbVie, Irvine, CA 92612, USA; ahunna.ukah@abbvie.com (A.U.); xiaomeng.yue@abbvie.com (E.Y.); kim.becker@abbvie.com (K.B.I.); norman.huang@abbvie.com (N.-Y.H.); 4Department of Neurology, University of California, Irvine, CA 92697, USA; swoped@hs.uci.edu

**Keywords:** cervical dystonia, botulinum neurotoxin, dysphagia, incidence, comorbidity, real-world evidence

## Abstract

Dysphagia is prevalent in patients with cervical dystonia (CD) and is a potential adverse event in patients treated with botulinum neurotoxin (BoNT) for CD. Real-world studies may provide a better understanding of the incidence and potential risk factors of dysphagia after BoNT administration. This retrospective cohort study used longitudinal patient-level data from Optum’s de-identified Market Clarity Data to evaluate rates of dysphagia in patients with CD exposed and unexposed to BoNT. Patients ≥18 years of age with ≥2 CD diagnoses ≥30 days apart during the study period (1 January 2017–30 September 2021) who had ≥180 days of continuous health plan enrollment prior to the cohort entry date (first CD diagnosis) were included. Overall, the mean (SD) age of all CD patients (Cohort 1; N = 81,884) was 54.00 (16.21) years, and they were mostly female (67.9%) and white (76.96%). BoNT-Exposed patients (Cohort 2; N = 19,244) had a higher incidence of dysphagia (16.3%) and comorbid conditions when compared with BoNT-Unexposed patients (Cohort 3; N = 61,154 [12.1%]). Overall, patients with pre-existing dysphagia, other dystonias, and comorbid neurologic conditions at baseline also had higher proportions of dysphagia. This real-world analysis indicates that comorbid conditions predispose patients with CD to a greater dysphagia risk during treatment.

## 1. Introduction

Cervical dystonia (CD) is the most common adult-onset focal dystonia [1]. It is a movement disorder characterized by the sustained or intermittent involuntary contraction of the neck and shoulder muscles, often resulting in repetitive abnormal neck movements or postures that may be tremulous [2,3]. For patients with CD, the primary goals of current therapeutic approaches are to ameliorate involuntary movements, correct abnormal postures, reduce pain, and improve overall function and quality of life [4]. While there are various treatment modalities available, chemodenervation of the affected muscles with botulinum neurotoxin (BoNT) injection is the accepted standard of care and is a first-line treatment for patients with CD [5,6,7].

In clinical trials, the efficacy and safety of BoNT for the treatment of CD has been well established [8,9]. After decades of use in the real world, BoNT has demonstrated continued effectiveness and a robust safety profile [10]. Nevertheless, dysphagia is prevalent among 11% to 68% of patients with CD [11,12,13,14] and is also a reported treatment-related adverse event (TRAE) in 2% to 25% of patients in registrational trials of marketed BoNT products [15,16,17,18]. Dysphagia was reported in 2% to 4% of patients in a more recent phase III trial with a new BoNT/A formulation [19], which is lower than the observed incidence in other pivotal registration trials [15,16,17,18]. However, these dysphagia rates should be interpreted with caution as an evaluation of the eligibility criteria indicated that patients with pre-existing grade ≥3 dysphagia and more complicated patients (i.e., significant dystonia in other areas of the body) were excluded [19]. In addition, patients with predominant retrocollis (RC) or anterocollis (AC) posture were also excluded [19]; AC is the most difficult CD posture to treat and patients with AC may be particularly at risk of dysphagia [20].

While the precise cause of dysphagia in patients with CD is not fully understood, studies suggest that dysphagia develops due to a functional deficit in swallowing from abnormalities in neck posture and a delayed stimulus in swallow initiation from neurogenic causes [11,14,21]. Dysphagia risk may be increased when utilizing BoNT therapy as well as with factors such as the injection site and total dose [22]. It is well understood that certain postures such as AC are at higher risk of developing dysphagia and require modifications to the dose and injection technique to mitigate the risk of dysphagia [23]. Patients with CD are a heterogenous patient population, and various factors may contribute to dysphagia onset. This retrospective study evaluated the real-world incidence of dysphagia in patients with CD by BoNT exposure, baseline patient characteristics, and other contributing factors that may lead to dysphagia that are often excluded from clinical trials.

## 2. Results

### 2.1. Patient Characteristics

A total of 81,884 patients with CD (Cohort 1, All CD Patients) met the study inclusion criteria; 98% (80,398/81,884) did not have a BoNT injection for the treatment of CD within 12 weeks prior to the cohort entry date (CED, Appendix A). Of these, 24% (19,244/80,398) had ≥1 BoNT injections (Cohort 2, BoNT-Exposed) and 76% (61,154/80,398) did not have BoNT injections (Cohort 3, BoNT-Unexposed) in the neck area on or after the CED.

The mean (SD) age of Cohort 1 (All CD Patients) at baseline was 54.00 (16.21) years, and they were mostly female (67.90%) and white (76.96%, Table 1). There were fewer patients in Cohort 2 (BoNT-Exposed) compared to Cohort 3 (BoNT-Unexposed: 19,244 vs. 61,154). Relative to Cohort 3 (BoNT-Unexposed), Cohort 2 (BoNT-Exposed) had more patients in older age groups (50–79 years) and had fewer males (34.20% vs. 25.92%) and African Americans (7.02% vs. 3.64%, Table 1).

Baseline characteristics were captured within 180 days prior to index date. Cohort 2 (BoNT-Exposed) had a greater proportion of patients with comorbidities when compared with Cohort 3 (BoNT-Unexposed). These included chronic migraine (21.2% vs. 9.6%) and other neurologic disorders that are known risk factors of dysphagia such as gastroesophageal reflux disease (16.68% vs. 15.24%) [24], stroke (5.77% vs. 4.64%) [25], and Parkinson’s disease (5.28% vs. 2.21%; Table 2) [26]. Predominant CD posture was not well documented for the majority of patients across cohorts.

### 2.2. Outcomes

The overall proportion of patients with dysphagia was 13.7% in Cohort 1 (All CD Patients, Figure 1). A higher proportion of patients with dysphagia was observed in Cohort 2 (BoNT-Exposed) compared with Cohort 3 (BoNT-Unexposed: 16.3% vs. 12.1%; Figure 1). Additionally, incidence rates (IRs) were 26.7 per 100 person-years (95% confidence intervals [CI]: 25.8–27.7) in Cohort 2 (BoNT-Exposed) and 5.6 per 100 person-years (95% CI: 5.5–5.8) in Cohort 3 (BoNT-Unexposed; Figure 1).

Dysphagia occurred more frequently in patients with pre-existing comorbidities compared to patients without these conditions at baseline (Figure 2). Key contributing baseline comorbidities included pre-existing dysphagia (Cohort 2 [BoNT-Exposed]: 63.7% vs. 12.9%; Cohort 3 [BoNT-Unexposed]: 61.7% vs. 9.8%), other dystonias (Cohort 2 [BoNT-Exposed]: 21.4% vs. 15.5%; Cohort 3 [BoNT-Unexposed]: 24.9% vs. 11.6%), and other neurologic conditions such as stroke, gastroesophageal reflux disease, and neuromuscular disorders (Cohort 2 [BoNT-Exposed]: 29.2% vs. 11.9%; Cohort 3 [BoNT-Unexposed]: 24.1% vs. 8.9%) (Figure 2).

The analysis of IRs (95% CI) demonstrated similar trends, indicating that patients with pre-existing comorbidities were more likely to experience dysphagia overall (Appendix A). Further, incident dysphagia rates were greatest after BoNT exposure and patients diagnosed with dysphagia during the baseline period were most likely to experience dysphagia following treatment (Cohort 2 [BoNT-Exposed]: dysphagia, 205.5 per 100 person-years [95% CI: 191.74–220.04]; other dystonia, 33.0 per 100 person-years [95% CI: 30.34–35.83]; other neurologic conditions, 53.9 per 100 person-years [95% CI: 51.19–56.78]; Appendix A). Patients with dysphagia had more hospitalizations and emergency room or urgent care visits relative to patients who did not develop dysphagia (Table 3).

In Cohort 2 (BoNT-Exposed), the mean (SD) time between a dysphagia diagnosis and the closest previous BoNT injection was 45.0 (79.7) days (Figure 3). Diagnosis of dysphagia occurred in the first 30 days after BoNT injection in 49% of patients. The observed increase in the proportion of patients with pre-existing dysphagia and development of dysphagia post-treatment were consistent across BoNT products (Appendix A). Patients in Cohort 2 (BoNT-Exposed) and Cohort 3 (BoNT-Unexposed) diagnosed with dysphagia during the follow-up period were older age (50–79 years) and had higher baseline comorbidity rates of gastroesophageal reflux disease and psychiatric comorbidities (Cohort 2 [BoNT-Exposed]: 27.39% and 38.68%; Cohort 3 [BoNT-Unexposed]: 29.09% and 36.50%, respectively (Table 3)).

Patients in Cohort 2 (BoNT-Exposed) that had continuous health plan enrollment of ≥12 months (73.4%) and ≥24 months (53.2%) received BoNT for a mean (range) of 3.0 (1.0–7.0) and 4.7 (1.0–14.0) injection cycles, respectively. Overall, onabotulinumtoxinA was the most frequently reported BoNT (88.2% [16,970/19,244]) with a mean (SD) dose of 185.8 U (35.8 (Figure 4)).

## 3. Discussion

In this retrospective analysis of real-world patient-level data from Market Clarity, the overall proportion of patients with dysphagia was 13.7% among patients with CD. A higher proportion of dysphagia cases was observed in patients with pre-existing neurologic comorbidities and baseline dysphagia across all cohorts during follow-up. Incidence rates of dysphagia were highest in patients exposed to BoNT with baseline comorbidities, suggesting that careful consideration of dose [27] and muscle selection for injection is advisable in these patients.

In this study, dysphagia cases were higher in Cohort 2 (BoNT-Exposed), at 16.3%, relative to patients unexposed to BoNT (Cohort 3, 12.1%). However, Cohort 2 (BoNT-Exposed) had a substantially smaller population size and included a higher proportion of older patients and patients with comorbidities which may signal a more severe disease burden and higher risk for the development of dysphagia compared with Cohort 3 (BoNT-Unexposed). In Cohort 2 (BoNT-Exposed), there was an average of 45 days between dysphagia diagnosis and BoNT treatment. Forty-nine percent of patients received a diagnosis of dysphagia within the first 30 days after BoNT injection and the remaining 51% were diagnosed ≥31 days post-BoNT exposure. Dysphagia is an adverse drug reaction associated with BoNT treatment from the local diffusion of BoNT into off-target muscles near the injection site and typically occurs within hours to weeks following treatment [15]. The occurrence of dysphagia after this 2–4-week period following BoNT injection may be caused by other contributing factors such as progressive neuromuscular changes, or the underlying disease state itself. Therefore, a direct cause-and-effect relationship between a dysphagia diagnosis and BoNT treatment cannot be fully established from this study.

Data on long-term use of BoNT therapy were not evaluated in this analysis, making it difficult to assess whether cumulative lifetime BoNT exposure was associated with dysphagia incidence exposure. Further, 12.1% of patients in Cohort 3 (BoNT-Unexposed) had dysphagia during the follow-up period, indicating that dysphagia is prevalent in patients with CD who are not exposed to BoNT. Lastly, the majority of patients received onabotA (88.2%), so comparisons of dysphagia rates between BoNT products were limited; however, no apparent correlations between the BoNT administered and the occurrence of dysphagia were observed.

Although data on the dose administered were not available for all patients, the proportion of patients with these data received BoNT doses close to the recommended individual product labeled dose, suggesting that patients received treatment in line with established recommendations [15,16,17,18]. A recent study analyzed dysphagia rates across multiple BoNT products’ pivotal studies and presented the hypothesis that the incidence of dysphagia is driven by core neurotoxin content [28]. However, in this large, real-world study that includes data from patients treated within an approximately 5-year time frame, the primary driver of dysphagia post-treatment with any BoNT was the occurrence of baseline dysphagia. Differences in dysphagia rates from pivotal studies are most likely due to differences in the clinical characteristics of patients included, the study design, and when these studies were conducted, as those performed more recently may include investigators with more injector experience than the pivotal trials performed decades ago rather than being determined by core neurotoxin content.

Previous real-world studies have demonstrated the safety of BoNT for the treatment of CD and indicate that dysphagia is a treatment-related AE in 6.4% to 19% of patients [10,27,29]. While our results are consistent with current real-world study estimates, the wide range of incidence indicates considerable variability, which may be attributed to underlying factors associated with dysphagia risk. For instance, dysphagia cases were observed in 12.9% of patients without comorbid baseline dysphagia in the BoNT-Exposed Cohort (Cohort 2), whereas a much greater proportion of patients in this cohort with baseline dysphagia were diagnosed with dysphagia during follow-up (63.7%). Dysphagia incidence was also associated with the presence of other comorbid dystonias or neurologic conditions at baseline. The rates of dysphagia reported herein further demonstrate that patients with pre-existing comorbid dysphagia were noticeably more likely to experience dysphagia following BoNT treatment. Pre-existing comorbid conditions may, therefore, predispose patients with CD to more frequent episodes of dysphagia and may be a risk factor for developing dysphagia following BoNT treatment. Consequently, rates of dysphagia reported in clinical trials may differ in patients with these underlying conditions, since they are typically excluded. Although assessments between dysphagia severity and pre-existing comorbidities were out of scope for this analysis, reports of dysphagia following BoNT treatment are typically mild to moderate [8,9,15].

Most patients’ predominant CD posture was captured as “unknown” in the current data set (Cohort 1 [All CD Patient]: 90.8%; Cohort 2 [BoNT]: 84.6%; Cohort 3 [BoNT-Unexposed]: 92.8%), so the analysis of dysphagia and contributing posture was limited. Nearly one third of patients (29.2%) exposed to BoNT therapy who reported an episode of dysphagia during the follow-up period had baseline neurological conditions that are known risk factors for dysphagia. Specifically, these patients presented with comorbid gastroesophageal reflux disease (16.68%), stroke (5.77%), and/or Parkinson’s disease (5.28%) at baseline. Among these patient populations, dysphagia is prevalent in up to 48% of patients with gastroesophageal reflux disease [24], 50% of stroke survivors [25], and 80% of patients with Parkinson’s disease [26]. The data reported herein provide a real-world assessment of dysphagia in a more heterogenous patient population that includes contributing risk factors.

Dysphagia symptoms may present for various reasons in patients with CD regardless of BoNT therapy [30]. Patients with CD commonly report difficulties with swallowing, coughing, and/or choking while eating, requiring a longer time to eat and needing to adjust their posture while eating [30]. Dysphagia can be clinically detected in up to 36% of patients with CD prior to any intervention, [14] suggesting that all patients with CD may have some risk of developing dysphagia. When initiating BoNT in CD patients, muscle selection, localization, and dose should be carefully considered to minimize the risk of dysphagia, as BoNT may be more likely to diffuse to surrounding muscles when utilizing higher doses or a larger volume [31]. Proper muscle localization is important when treating conditions in the head and neck area with BoNT. A recent study on the use of ultrasound when injecting BoNT into the submandibular glands identified optimal injection sites based on external anatomical landmarks and key injection guidelines determined by electrophysiological examinations that could attenuate potential adverse events such as dysphagia and xerostomia with BoNT treatment [32]. In patients with CD, the differentiation of CD subtypes via the phenomenological classification of dystonia known as the collis–caput (COL-CAP) concept [33] could lead to a more accurate identification of target muscles for injection sites and, when coupled with assisted methods of BoNT delivery (e.g., electromyography and ultrasound), may mitigate adverse effects and improve efficacy [34]. An awareness of anatomical considerations and minimizing the extent of unintended BoNT diffusion with improved injection techniques may, therefore, reduce dysphagia rates in CD patients.

This analysis has limitations. This study was retrospective in nature and was limited to patients with continuous healthcare coverage. In addition, outcomes defined by diagnostic codes may be subject to measurement error (i.e., limited sensitivity or specificity) or lead to a misdiagnosis. Patient data including the efficacy and severity of CD were unable to be collected and analyzed. Further, limited data were captured on predominant CD posture, and clinician data, including BoNT injection experience and localization techniques (i.e., electromyography, ultrasound), were unable to be collected. Nonetheless, this study is the first of its kind to analyze over 81,000 patients with CD in a real-world setting, leading to a better understanding of the diverse nature of this patient population and the association of dysphagia with multiple variables. The breadth of patient characteristics included and the observed trends in this descriptive analysis provides a source of hypotheses to be tested more rigorously. 

Future research to develop a clinical prediction model using logistic regression modeling may improve patient risk stratification [35]. Variables such as pre-existing dysphagia and concurrent neurological conditions from this study, as well as additional data on dose and muscle selection, could assist in generating a predictive model in which patients with CD are stratified by dysphagia risk more accurately, leading to a more tailored treatment approach to better inform healthcare professionals that treat CD on clinical decision making.

## 4. Conclusions

This real-world study analyzed data from a large, heterogeneous population of patients with CD. The incidence of dysphagia among these patients was related to risk factors including baseline dysphagia, comorbidities including other dystonias, and other neurologic conditions such as stroke, gastroesophageal reflux disease, neuromuscular disorders, and treatment with BoNT. These comorbidities may predispose patients with CD to a higher incidence of dysphagia.

## 5. Materials and Methods

### 5.1. Study Design and Participants

This retrospective cohort study used longitudinal patient-level data from Optum’s de-identified Market Clarity Data (Market Clarity), which links medical and pharmacy claims with electronic health record data. Data were collected during the study period between 1 January 2017 and 30 September 2021. 

Eligible patients were ≥18 years of age on the CED, defined as the date of first CD diagnosis claim during the study period; had at least 2 diagnoses of CD per the International Classification of Diseases, Tenth Revision, Clinical Modification (ICD-10-CM, Appendix A) separated by ≥30 days; and had ≥180 days of continuous health plan enrollment prior to the CED with a gap of <45 days allowed (baseline period). All patients with CD who met the above inclusion criteria were categorized as Cohort 1 (All CD Patients). Cohorts 2 and 3 were mutually exclusive and included patients who were administered any BoNT (BoNT-Exposed) and administered no BoNT (BoNT-Unexposed) during the study period, respectively. Patients with any BoNT treatment in the neck area within 12 weeks prior to the CED were excluded from Cohorts 2 and 3 only. Index dates were defined as the day of first CD diagnosis (Cohorts 1 and 3) or start of BoNT therapy (Cohort 2). For Cohort 2, BoNT exposure was defined as the injection of any BoNT in the neck area from injection date plus 16 weeks (12 weeks recommended dosing interval plus 4 weeks); reinjection of any BoNT product before the end of the 16-week dosing interval resulted in the restarting of another 16-week dosing interval period.

### 5.2. Outcomes

The primary outcome of this analysis was dysphagia incidence, defined as any inpatient or outpatient diagnosis of dysphagia per the ICD-10-CM (Appendix A) during follow-up (after CED).

### 5.3. Statistical Analysis

Baseline characteristics were summarized using descriptive statistics and reported as the mean (SD) or frequency (%). Patients could have multiple comorbidities at once and were counted separately for each comorbidity. IRs and 95% CI of dysphagia were calculated for each cohort as the total incidence of dysphagia diagnoses divided by total person-years of follow-up and expressed as the number of events/100 person-years of follow-up. Each patient could only contribute one event of dysphagia to this calculation. Patient-time during follow-up was defined as the date of first CD diagnosis in Cohort 1, BoNT injection in Cohort 2, and first CD diagnosis in Cohort 3 to an incident of dysphagia, end of BoNT exposure (16 weeks), end of data availability, disenrollment, or death, whichever occurred first (Figure 5). For individual BoNT injections, the dates of switching to another BoNT and end of BoNT exposure were included as end-determinants of patient-time in Cohort 2.

## Figures and Tables

**Figure 1 toxins-17-00148-f001:**
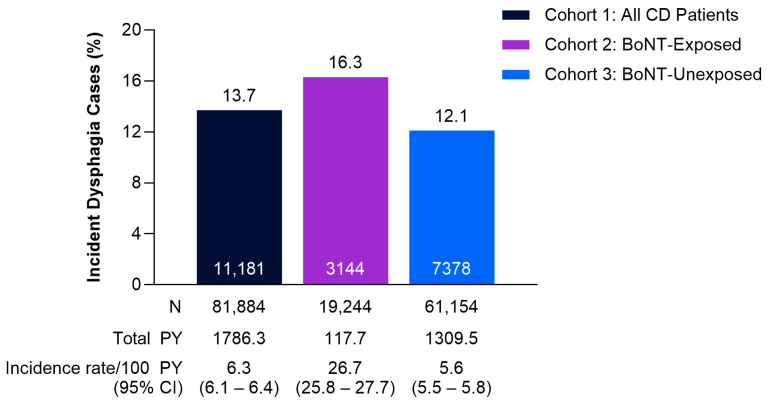
Proportion of patients with dysphagia and incident dysphagia rates per 100 person-years by cohort. Values within data bars (white text) indicate actual number of total cases. BoNT, botulinum neurotoxin; CD, cervical dystonia; CI, confidence interval; PY, person-years.

**Figure 2 toxins-17-00148-f002:**
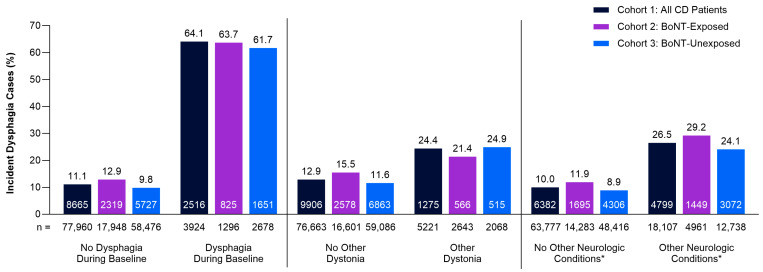
Proportion of patients with dysphagia and presence or absence of baseline comorbidities by cohort. * Neurologic conditions that are known risk factors for dysphagia include stroke, Parkinson’s disease, multiple sclerosis, Huntington’s disease, Wilson’s disease, gastroesophageal reflux disease, and other neuromuscular disorders. Values within data bars (white text) indicate actual number of total cases of dysphagia. BoNT, botulinum neurotoxin; CD, cervical dystonia.

**Figure 3 toxins-17-00148-f003:**
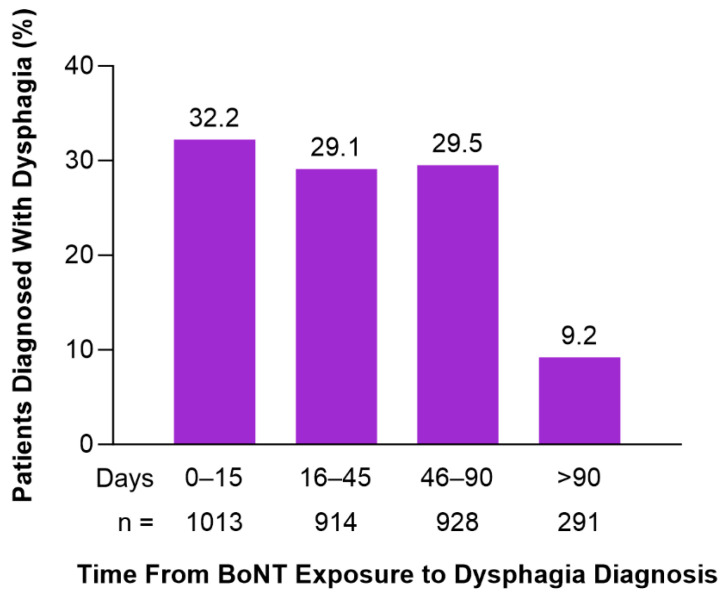
Time to dysphagia diagnosis from the most recent previous BoNT injection among patients previously exposed to BoNT treatments (Cohort 2) and diagnosed with dysphagia (N = 3144).

**Figure 4 toxins-17-00148-f004:**
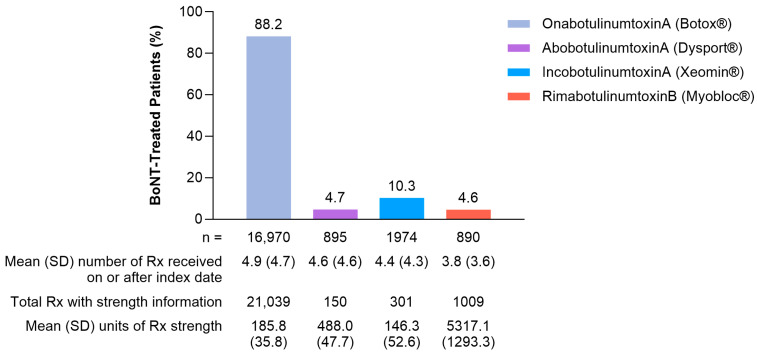
BoNT dose information among patients previously exposed to BoNT treatments (Cohort 2, N = 19,244). BoNT, botulinum neurotoxin; Rx, prescriptions; SD, standard deviation.

**Figure 5 toxins-17-00148-f005:**
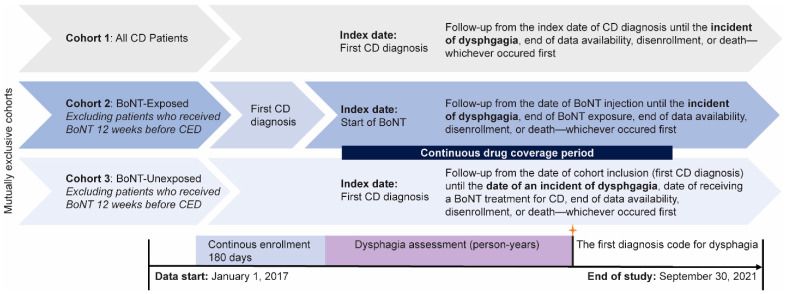
Study design and corresponding cohort patient-time description. BoNT, botulinum neurotoxin; CD, cervical dystonia; CED, cohort entry date.

**Table 1 toxins-17-00148-t001:** Baseline demographics.

Characteristic	Cohort 1: All CD Patients (N = 81,884)	Cohort 2: BoNT-Exposed (N = 19,244)	Cohort 3: BoNT-Unexposed (N = 61,154)
Mean (SD) age, years	54.00 (16.21)	56.92 (14.92)	53.10 (16.55)
Age group, years, n (%)			
18–29	6496 (7.93)	809 (4.20)	5580 (9.12)
30–39	10,893 (13.30)	1869 (9.71)	8851 (14.47)
40–49	14,122 (17.25)	3163 (16.44)	10,643 (17.40)
50–59	18,472 (22.56)	4617 (23.99)	13,388 (21.89)
60–69	16,783 (20.50)	4671 (24.27)	11,853 (19.38)
70–79	10,411 (12.71)	2977 (15.47)	7,286 (11.91)
≥80	4707 (5.75)	1138 (5.91)	3553 (5.81)
Female, n (%)	55,602 (67.90)	14,249 (74.04)	40,240 (65.80)
Race, n (%)			
White	63,014 (76.96)	15,695 (81.56)	46,112 (75.40)
African American	5032 (6.15)	701 (3.64)	4296 (7.02)
Asian	1215 (1.48)	231 (1.20)	974 (1.59)
Other/Unknown	12,623 (15.42)	2617 (13.60)	9772 (15.98)

BoNT, botulinum neurotoxin; CD, cervical dystonia; SD, standard deviation.

**Table 2 toxins-17-00148-t002:** Baseline clinical characteristics.

Characteristic	Cohort 1: All CD Patients (N = 81,884)	Cohort 2: BoNT-Exposed (N = 19,244)	Cohort 3: BoNT-Unexposed (N = 61,154)
Predominant CD posture, n (%)			
Laterocollis	1470 (1.8)	1027 (5.34)	391 (0.64)
Torticollis	7171 (8.76)	2703 (14.05)	4260 (6.97)
Unknown	74,308 (90.75)	16,277 (84.58)	56,766 (92.82)
Charlson Comorbidity Index, mean (SD)	1.3 (2.1)	1.4 (2.0)	1.3 (2.1)
Comorbidities, within 180 days prior to index date, n (%)			
Other types of dystonia	5221 (6.38)	2643 (13.73)	2068 (3.38)
Chronic migraine ^a,b^	10,424 (12.7)	4087 (21.2)	5918 (9.6)
Neurological conditions that are known risk factors for dysphagia, n (%)			
Gastroesophageal reflux disease	12,796 (15.63)	3209 (16.68)	9321 (15.24)
Stroke	4030 (4.92)	1110 (5.77)	2838 (4.64)
Parkinson’s disease	2433 (2.97)	1016 (5.28)	1354 (2.21)
Multiple sclerosis	867 (1.06)	316 (1.64)	517 (0.85)
Myasthenia gravis	109 (0.13)	20 (0.10)	86 (0.14)
Huntington disease	33 (0.04)	13 (0.07)	18 (0.03)
Wilson’s disease	11 (0.01)	7 (0.04)	4 (0.01)
Lambert–Eaton syndrome	3 (<0.01)	2 (0.01)	1 (<0.01)
Psychiatric comorbidities, n (%)	22,997 (28.08)	6207 (32.25)	22,997 (28.08)
Anxiety	17,028 (20.80)	4497 (23.37)	12,151 (19.87)
Depression	14,062 (17.17)	3993 (20.75)	9711 (15.88)

^a^ Number of patients receiving BoNT for treating migraine among patients with CD and chronic migraine: Cohort 1, 2132/4221 (50.5%); Cohort 2, 916/1621 (56.5%); Cohort 3, 968/2344 (41.3%). ^b^ Number of patients with CD and at least one diagnosis of chronic migraine at any time prior to index date. BoNT, botulinum neurotoxin; CD, cervical dystonia; SD, standard deviation.

**Table 3 toxins-17-00148-t003:** Baseline characteristics by dysphagia incidence during follow-up.

	Cohort 2: BoNT-Exposed (N = 19,244)		Cohort 3: BoNT-Unexposed (N = 61,154)	
Characteristic, n (%)	With Dysphagia (n = 3144)	Without Dysphagia (n = 16,100)	With Dysphagia (n = 7378)	Without Dysphagia (n = 53,776)
Mean (SD) age, years	60.43 (15.09)	56.23 (14.71)	59.27 (16.12)	52.25 (16.43)
Age group, years				
18–29	131 (4.17)	678 (4.21)	351 (4.76)	5229 (9.72)
30–39	181 (5.76)	1688 (10.48)	629 (8.53)	8222 (15.29)
40–49	391 (12.44)	2772 (17.22)	969 (13.13)	9674 (17.99)
50–59	691 (21.98)	3926 (24.39)	1562 (21.17)	11,826 (21.99)
60–69	779 (24.78)	3892 (24.17)	1697 (23.00)	10,156 (18.89)
70–79	675 (21.47)	2302 (14.30)	1373 (18.61)	5913 (11.00)
≥80	296 (9.41)	842 (5.23)	797 (10.80)	2756 (5.12)
Other types of dystonia	566 (18.00)	2077 (12.90)	515 (6.98)	1553 (2.89)
Stroke	370 (11.77)	740 (4.60)	781 (10.59)	2057 (3.83)
Parkinson’s disease	444 (14.12)	572 (3.55)	527 (7.14)	827 (1.54)
Gastroesophageal reflux disease	861 (27.39)	2348 (14.58)	2146 (29.09)	7175 (13.34)
Psychiatric comorbidities (of the below)	1216 (38.68)	4991 (31.00)	2693 (36.50)	13,556 (25.21)
Anxiety	853 (27.13)	3644 (22.63)	1979 (26.82)	10,172 (18.92)
Depression	856 (27.23)	3137 (19.48)	1742 (23.61)	7969 (14.82)
Hospitalization/emergency room visit/urgent care visit within 180 days prior to index date				
Patients with ≥1 hospitalization	522 (16.60)	1215 (7.55)	1485 (20.13)	4730 (8.80)
Patients with ≥emergency room or urgent care visit	954 (30.34)	3308 (20.55)	2616 (35.46)	13,131 (24.42)

BoNT, botulinum neurotoxin.

## Data Availability

AbbVie is committed to responsible data sharing regarding the clinical trials we sponsor. This includes access to anonymized, individual, and trial-level data (analysis data sets), as well as other information (e.g., protocols, clinical study reports, or analysis plans), as long as the trials are not part of an ongoing or planned regulatory submission. This includes requests for clinical trial data for unlicensed products and indications. These clinical trial data can be requested by any qualified researchers who engage in rigorous, independent, scientific research, and will be provided following review and approval of a research proposal, Statistical Analysis Plan (SAP), and execution of a Data Sharing Agreement (DSA). Data requests can be submitted at any time after approval in the US and Europe and after acceptance of this manuscript for publication. The data will be accessible for 12 months, with possible extensions considered. For more information on the process or to submit a request, visit the following link: https://vivli.org/ourmember/abbvie/ (accessed on 14 March 2025) then select “Home”.

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
