# Peer review of "Incidence of Dysphagia and Comorbidities in Patients with Cervical Dystonia, Analyzed by Botulinum Neurotoxin Treatment Exposure"

_toxins, 2025, doi:10.3390/toxins17030148_

Round 1

Reviewer 1 Report

Comments and Suggestions for Authors

Overall Assessment:
The manuscript presents an important real-world analysis of dysphagia incidence in patients with cervical dystonia (CD) following botulinum neurotoxin (BoNT) treatment. The study leverages a large dataset and highlights significant findings regarding comorbidities, injection-related risks, and patient outcomes. However, there are key areas that require clarification, additional references, and further discussion to improve the manuscript’s comprehensiveness and clinical impact.

Major Comments:

  1. Injection Site and Dysphagia Risk:
    The manuscript discusses the risk of dysphagia following BoNT treatment but does not provide sufficient details on the role of injection site selection. There is evidence that BoNT injections into the submandibular gland can contribute to dysphagia by impairing saliva production and swallowing mechanics. I strongly recommend citing the following study to support this point:

    Yi KH, Kim SB, Hu H, An HS, Hidajat IJ, Lim TS, Kim HJ. Ultrasonographic Study of the Submandibular Gland for Botulinum Neurotoxin Injection. Dermatol Surg. 2024 Sep 1;50(9):834-837. doi: 10.1097/DSS.0000000000004208. Epub 2024 May 7. PMID: 38712848.

    This study provides critical insights into anatomical considerations for BoNT injection and highlights how unintended diffusion or incorrect targeting can contribute to dysphagia. A discussion of these findings would strengthen the paper’s discussion on injection technique optimization.

  2. Temporal Relationship Between BoNT Injection and Dysphagia:
    The study reports a mean of 45 days between BoNT injection and dysphagia diagnosis, with 49% of cases occurring within 30 days. This observation warrants further discussion on whether early-onset cases may be due to diffusion effects, while later-onset cases could be related to cumulative dosing or progressive neuromuscular changes. Clarification of these mechanisms, possibly with reference to imaging-based studies on toxin spread, would enhance the discussion.

  3. Dose-Response Relationship and Injection Guidance:
    While the manuscript states that "BoNT dose, muscle selection, number of treatments, and localization technique... are not as strongly linked with the development of dysphagia," this contradicts some existing literature indicating that precise targeting using ultrasound or electromyography (EMG) can mitigate adverse effects. Additional discussion on the role of advanced localization techniques would improve the manuscript’s practical recommendations.

  4. Comorbidity Stratification and Risk Prediction Models:
    The study provides valuable data on comorbidities predisposing patients to dysphagia but lacks a risk stratification model. Future research directions should include predictive modeling incorporating variables such as injection location, pre-existing dysphagia, and concurrent neurologic conditions. A brief discussion on how such models could inform clinical practice would be beneficial.

Minor Comments:

  • Table 1: Adding a column for "time since CD diagnosis" could provide additional insights into whether disease duration influences dysphagia risk.
  • Figures 1 & 2: Clarifying whether incident dysphagia rates were adjusted for baseline differences between BoNT-exposed and unexposed groups would improve interpretability.
  • Discussion Section: Consider briefly addressing potential treatment modifications (e.g., dose titration, injection site avoidance) that could minimize dysphagia risk.
Comments on the Quality of English Language

N/A

Reviewer 2 Report

Comments and Suggestions for Authors

This paper investigated the potential risk of dysphagia in patients with cervical dystonia with/without botulinum neurotoxin treatment. It included a large number of patients, over 80,000, and compared the incidence rate of dysphagia between BoNT-exposed and BoNT-unexposed groups. The results and discussion are interesting with regard to predisposing risk of dysphagia during follow-up of CD, highlighting that pre-existing dysphagia or comorbidities are important indicators.

My comments are:  

1) Check the numbers of patients in Tables 1 and 2. For example, the total number of patients in Cohort 2 and Cohort 3 does not match the number of patients in Cohort (all) in each age group in Table 1.  

2) CED and IRs first appeared on pages 2 and 3, respectively. However, their spelling appeared on page 9. They should be shown at their first appearance.  

Comments on the Quality of English Language

-

Round 2

Reviewer 1 Report

Comments and Suggestions for Authors

The comments are all in the revised text.

They seems to be acceptable.